# Targeting EZH2 histone methyltransferase activity alleviates experimental intestinal inflammation

Jie Zhou[1,7], Shuo Huang[1,7], Zhongyu Wang[1,7], Jiani Huang[1], Liang Xu[1], Xuefeng Tang[2], Yisong Y. Wan[3,4], Qi-jing Li[5], Alistair L.J. Symonds[6], Haixia Long[1] & Bo Zhu[1]

Enhancer of zeste homolog 2 (EZH2)-mediated trimethylation of histone 3 lysine 27 (H3K27Me3) is critical for immune regulation. However, evidence is lacking to address the effect of EZH2 enzyme's activity on intestinal immune responses during inflammatory bowel disease (IBD). Here we report that suppressing EZH2 activity ameliorates experimental intestinal inflammation and delayed the onset of colitis-associated cancer. In addition, we identified an increased number of functional MDSCs in the colons, which are essential for EZH2 inhibitor activity. Moreover, inhibition of EZH2 activity promotes the generation of MDSCs from hematopoietic progenitor cells in vitro, demonstrating a previously unappreciated role for EZH2 in the development of MDSCs. Together, these findings suggest the feasibility of EZH2 inhibitor clinical trials for the control of IBD. In addition, this study identifies MDSC-promoting effects of EZH2 inhibitors that may be undesirable in other therapeutic contexts and should be addressed in a clinical trial setting.

[1] Institute of Cancer, Xinqiao Hospital, Third Military Medical University, Chongqing 400037, China. [2] Department of Pathology, Xinqiao Hospital, Third Military Medical University, Chongqing 400037, China. [3] Department of Microbiology and Immunology, School of Medicine, University of North Carolina at Chapel Hill, Chapel Hill, North Carolina 27599, USA. [4] Lineberger Comprehensive Cancer Center, University of North Carolina at Chapel Hill, Chapel Hill, North Carolina 27599, USA. [5] Department of Immunology, Duke University Medical Center, Durham 27710 North Carolina, USA. [6] Institute of Cell and Molecular Science, Barts and London School of Medicine and Dentistry, University of London, London E1 2AT, UK. [7] These authors contributed equally: Jie Zhou, Shuo Huang, Zhongyu Wang. Correspondence and requests for materials should be addressed to H.L. (email: longhaixialhx@163.com) or to B.Z. (email: bo.zhu@tmmu.edu.cn)

Inflammatory bowel disease (IBD), including ulcerative colitis and Crohn's disease, is characterized by chronic, relapsing gastrointestinal inflammation. IBD is a world-wide health problem with sharply increasing prevalence among all populations[1]. Patients with IBD suffer poor quality of life and are at a two- to eightfold higher risk of colorectal cancer (termed colitis-associated colorectal cancer, CAC)[2]. However, current drug therapies for IBD are far from optimal[3]: the efficacy of amino-salicylates is modest; steroids are effective but cause severe complications; and biologic therapies such as anti-tumor necrosis factor (TNF)-α antibody are associated with increased opportunistic infections in some patients[4]. Thus, new safe and efficacious therapies need to be developed.

The key characteristic of IBD is unresolved inflammation in the intestinal tract, which is caused by a failure to switch from a pro-inflammatory response to an anti-inflammatory response[5,6]. Multiple immune cells populations, including macrophages, dendritic cells (DCs), neutrophils, and eosinophils present in the intestinal mucosa contribute to inflammation during IBD. These cell types initiate inflammatory responses by secreting anti-microbial agents, chemokines, and pro-inflammatory cytokines including interleukin (IL)-6 and IL-1β. In contrast, regulatory T (Treg) cells impose immunosuppression to restrict IBD, such that Treg transfer abrogates the development of experimental colitis[7]. Myeloid-derived suppressor cells (MDSCs) exert immunosuppressive effects on both innate and adaptive immune systems to foster immune tolerance. Similarly, the adoptive transfer of MDSCs in different animal models of IBD ameliorates colitis[8–10]. Hence, promoting the accumulation of anti-inflammatory cells may be effective to treat IBD.

Intestinal immune homeostasis is controlled by epigenetic histone-modifying factors, including histone deacetylases and methyltransferases[11]. Aberrant histone modifications are present in IBD and are believed to be important for IBD pathogenesis[12,13]. Thus, interfering with the function of histone regulators may be a beneficial approach to the development of novel therapeutic strategies. In fact, accumulating evidence suggests that inhibition of histone deacetylation can reduce experimental IBD by promoting immune tolerance[14,15]. Enhancer of zeste homolog 2 (EZH2), a major histone methyltransferase, plays an essential role in immune regulation via trimethylating lysine 27 on histone H3. EZH2 has several known functions in immunity. First, EZH2 promotes the self-renewal of hematopoietic stem cells[16]. Second, EZH2 regulates the differentiation of hematopoietic stem and progenitor cells, and further controls immune cell lineage development, including T, B, and natural killer cell lineages[17,18]. Third, both innate and adaptive immune responses are also modulated by EZH2 via promotion of macrophage M1 polarization[19], regulation of the differentiation and responses of Th1, Th2, and Th17 cells, and maintenance of Treg identity[20,21]. Given the known role of immune cells in IBD pathogenesis and the involvement of EZH2 in immune cell development, differentiation, and function, we hypothesized that interfering with EZH2 activity may alter IBD progression by influencing the intestinal immune response.

Here we report that interference with EZH2 activity using pharmacological inhibitors attenuates dextran sodium sulfate (DSS)-induced colitis and delays the onset of CAC. Using GSK343, a selective EZH2 inhibitor, we demonstrate that the protective role of EZH2 inhibition is associated with increased MDSC populations in the colonic lamina propria (cLP). Notably, inhibition of EZH2 activity also promotes the generation of MDSCs from hematopoietic progenitor cells (HPCs). Together, our data identify EZH2 activity inhibition as a promising therapeutic approach to treat IBD. Furthermore, these results also warn against using EZH2 inhibitors for treating cancer clinically, as they may suppress beneficial anticancer immunity responses by increasing MDSC populations.

## Results

**Inhibition of EZH2 activity delays onset of colitis.** To investigate the impact of targeting EZH2 methyltransferase activity on the development of IBD, we first treated healthy C57BL/6 mice with GSK343 and GSK126 (both compounds compete with S-adenosyl-methionine for binding to EZH2, thereby inhibiting histone methyltransferase activity without affecting EZH2 protein expression). Over 11 days, no evident clinical or histologic abnormalities were observed (Supplementary Fig. 1), indicating that both drugs in current solution and dosing were not toxic. We next challenged C57BL/6 mice with water-fed DSS, a widely used chemical irritant that induces intestinal inflammation with the clinical, immunological, and histological features of human IBD[22]. Vehicle (control) or GSK343 was intravenously (i.v.) injected at different time points during the course of colitis development (Fig. 1a). We found that preventive GSK343 treatment significantly attenuated weight loss caused by DSS (Fig. 1b). In addition, the disease activity index (DAI), a composite score used to evaluate the clinical manifestations of colitis, was also decreased upon GSK343 treatment (Fig. 1c). Reduced colon length often serves as a surrogate macroscopic indication of colonic injury and we observed significantly longer colons in GSK343-treated mice than in the vehicle-treated group following DSS exposure (Fig. 1d). These findings suggest that treatment with GSK343 reduces the onset and clinical signs of DSS-induced colitis.

Detailed histological analysis of colonic lesions from vehicle-treated colitic mice showed severe pathology, as evidenced by a widely disrupted crypt architecture and abundant inflammatory cell infiltration. In contrast, GSK343 treatment markedly reduced these pathological changes, with a well-preserved mucosal architecture, limited inflammation, and small foci of crypt loss, leading to a decreased histological score (Fig. 1e). As the excessive secretion of pro-inflammatory cytokines is closely associated with intestinal inflammation and IBD clinical symptoms[23], we next assessed whether GSK343 treatment affects pro-inflammatory cytokine production. To this end, we measured the concentrations of IL-6, IL-1β, TNF-α, and IL-17A in colonic tissues at day 7 after DSS induction. Indeed, GSK343 administration led to a profound decrease in the levels of pro-inflammatory cytokines, including IL-6 and IL-1β, whereas the levels of TNF-α and IL-17A remained unaffected, indicating that GSK343 treatment reduces the inflammatory response during DSS-induced colitis (Fig. 1f). Upon increasing the DSS concentration to 3%, a dose at which ~70% of the vehicle-treated mice died due to acute colitis, we observed that only 20% of the GSK343-treated mice succumbed to colitis (Fig. 1g).

We then wondered whether another EZH2 inhibitor, GSK126, which is currently being evaluated in clinical trials to treat cancer (ClinicalTrials.gov identifier: NCT02082977)[24], can exert effects on DSS-induced colitis similar to those observed with GSK343. Indeed, treatment with GSK126 also significantly mitigated symptoms of DSS-induced colitis including weight loss, DAI, colonic shortening, pathology, and death (Supplementary Fig. 2).

Having identified an essential role for EZH2 in DSS-induced colitis, we employed a second intestinal inflammation model, indomethacin-induced enteropathy, to substantiate and expand on this idea. This mouse model mimics the clinical pathology of nonsteroidal anti-inflammatory drug-induced enteropathy in humans, a common intestinal disease with high mortality and a low cure rate[25,26]. Similar to our observations for the DSS model, GSK343 treatment significantly relieved indomethacin-induced

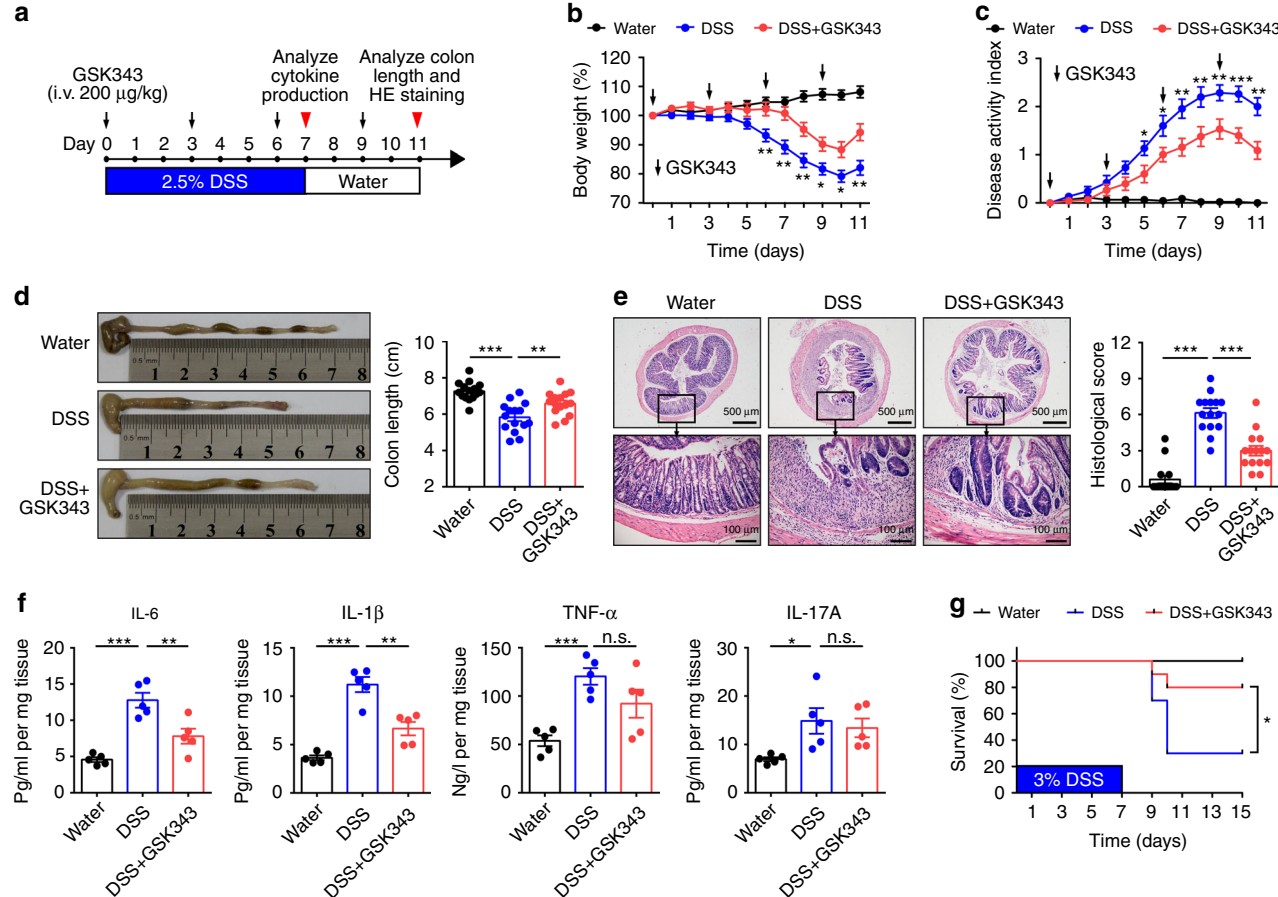

**Fig. 1** Inhibition of EZH2 activity preventively ameliorates DSS colitis. **a** Methods for DSS-induced acute colitis in C57BL/6 mice and GSK343 administration. **b**, **c** Body weight (**b**) and disease activity index (**c**) of mice that received regular drinking water alone (Water group, black dotted line) or 2.5% DSS-containing water (DSS group, blue dotted line), or 2.5% DSS combined with GSK343 injection (DSS + GSK343 group, red dotted line) ($n = 15$ per group). For statistical comparisons, asterisk indicates DSS vs. DSS + GSK343. **d**, **e** Colon length (**d**), representative hematoxylin and eosin (H&E) staining of distal colon sections, and corresponding histological scores (**e**) at day 11 after DSS induction ($n = 15$ per group). **f** Colonic inflammatory cytokine analysis from distal colonic tissue at day 7 after DSS induction ($n = 5$ per group). **g** Survival curves of mice treated with drinking water, 3% DSS, or 3% DSS plus GSK343 ($n = 10$ per group). Data are representative of three independent experiments. *$P < 0.05$ by log-rank test. **b–f** Data are representative of three independent experiments. Quantitative data are shown as the mean ± SEM. The statistical significance of differences was determined by two-way analysis of variance with Bonferroni post test (**b**, **c**) and one-way analysis of variance followed by Bonferroni post test (**d–f**). *$P < 0.05$, **$P < 0.01$, ***$P < 0.001$, n.s. = not significant. Source data are provided as a Source Data file

enteropathy, as evidenced by decreased weight loss, intestinal shortening, and histological damage (Supplementary Fig. 3).

Taken together, these results demonstrate that inhibition of EZH2 methyltransferase activity has a potent preventive effect on intestinal inflammation.

**Inhibition of EZH2 activity alleviates ongoing colitis**. To further address whether inhibition of EZH2 can ameliorate ongoing DSS-induced colitis, we therapeutically treated DSS-induced mice with GSK343. GSK343 was administrated when more than half of DSS-treated mice presented colitis symptoms (soft or loose stools, diarrhea, visual pellet bleeding, or grossly bloody or dark stools) (Fig. 2a). Although vehicle- and GSK343-treated mice lost weight similarly (Fig. 2b), GSK343 treatment markedly improved the clinical symptoms. Overall, DSS-inflicted mice that received vehicle treatment exhibited much more severe colitis symptoms, including severe mental malaise, loose or bloody stools that stuck to the anus, abundant bloody diarrhea, and a higher DAI score compared with those who received GSK343 treatment (Fig. 2c). Moreover, other signs of colitis, including colonic shortening (Fig. 2d), pathology (Fig. 2e), and death (Fig. 2f), were

significantly improved after GSK343 treatment. Together, these results show that EZH2 methyltransferase inhibition also provides therapeutic benefits for DSS-induced colitis.

IBD is an independent risk factor for developing CAC and the severity of colitis is directly linked to CAC in the azoxymethane (AOM)/DSS model[27]. Therefore, we reasoned that inhibition of EZH2 activity during colitis development may affect the development of CAC. To test this hypothesis, we combined the carcinogen AOM with two cycles of DSS-induced colitis to trigger CAC and then treated mice with vehicle or GSK343 when more than half of mice developed colitis (Supplementary Fig. 4a). In agreement with the aforementioned results, GSK343 treatment led to a decreased DAI score compared with control during colitis (Supplementary Fig. 4b). By day 72, whereas 100% of vehicle-treated mice developed CAC, only 50% of GSK343-treated mice did, and they had a reduced tumor burden (Supplementary Fig. 4c–e). By day 84, although similar percentages of vehicle- and GSK343-treated mice developed CAC, GSK343-treated mice had a significantly reduced tumor burden (Supplementary Fig. 4f-h). Therefore, inhibition of EZH2 activity during colitis delays the onset of CAC.

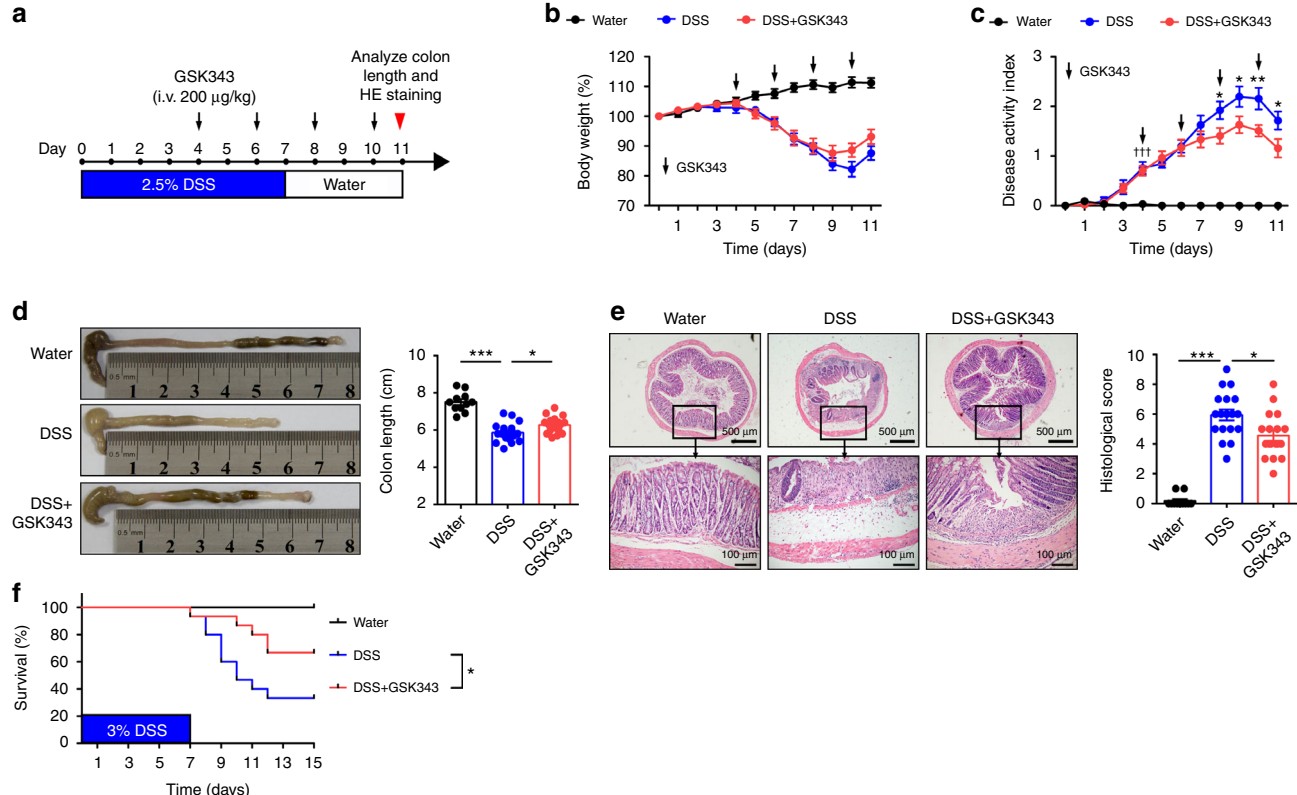

**Fig. 2** Therapeutic inhibition of EZH2 activity alleviates DSS colitis. **a** The experimental design of the therapeutic treatment with GSK343 on DSS-induced acute colitis in C57BL/6 mice. **b**, **c** Body weight (**b**) and disease activity index (**c**) ($n = 11$, Water; $n = 18$ for both DSS and DSS + GSK343) of mice that received regular drinking water alone (Water group, black dotted line) or 2.5% DSS (DSS group, blue dotted line), or 2.5% DSS combined with GSK343 injection (DSS + GSK343 group, red dotted line). For statistical comparisons, dagger indicates DSS vs. Water and asterisk indicates DSS vs. DSS + GSK343. **d**, **e** Colon length (**d**), representative H&E image of distal colon sections and corresponding histological scores (**e**) at day 11 after DSS exposure ($n = 11$, Water; $n = 18$ for both DSS and DSS + GSK343). **f** Survival curves of the indicated mice ($n = 15$ per group). Data are representative of three independent experiments. *$P < 0.05$ by log-rank test. **b**–**e** Data are representative of three independent experiments. The statistical significance of differences was determined by two-way analysis of variance with Bonferroni post-test (**b**, **c**), and one-way analysis of variance followed by Bonferroni post-test (**d**, **e**). *$P < 0.05$, **$P < 0.01$, ***$P < 0.001$, †††$P < 0.001$. Error bars indicate means ± SEM. Source data are provided as a Source Data file

**MDSC is critical to ameliorate experimental colitis**. Given the crucial role of immune mechanisms in the pathogenesis of IBD as well as the regulatory functions of EZH2 on both innate and adaptive immune cells, we next investigated whether and how GSK343 treatment impacts colonic immune cell infiltrates. Over time, adaptive immune disorders including Th17/Treg transformation imbalance are thought to be the main cause of IBD pathogenesis. However, comparable proportions and absolute numbers of IL-17A-expressing CD4+ T cells (Th17) and Foxp3-expressing CD4+ T cells (Tregs) were found in the cLP of DSS-exposed mice regardless of GSK343 treatment (Supplementary Fig. 5a), indicating that both Th17 and Treg cells may not be critical for GSK343-mediated amelioration of DSS-induced colitis.

A recent study reported that systemic inhibition of EZH2 protein expression by DZNep results in heightened susceptibility to DSS-induced colitis, which is associated with an activated, effector phenotype of Treg cells[28]. Interestingly, our data demonstrate that systemic inhibition of EZH2 histone methyltransferase activity by GSK343 ameliorates DSS colitis. To reconcile these different observations, we tested the expression of cell-surface markers in Treg cells. Notably, we failed to detect any significant differences in the expression of these cell-surface markers in Treg cells between GSK343-treated and vehicle-treated colitic mice (Supplementary Fig. 5b), suggesting a distinct mechanism by which GSK343 regulated the progress of colitis. To

further evaluate whether adaptive immune cells are required to mediate the beneficial effects of GSK343 on colitis, we induced DSS colitis in non-obese diabetes/severe combined immunodeficiency (NOD/SCID) mice, which lack adaptive immunity, and prophylactically treated them with GSK343 (Supplementary Fig. 5c). Similar to the results in wild-type C57BL/6 mice, GSK343-treated NOD/SCID mice showed reduced signs of colitis, including weight loss (Supplementary Fig. 5d), DAI scores (Supplementary Fig. 5e), colon shortening (Supplementary Fig. 5f), and pathology (Supplementary Fig. 5g). These findings suggest that the beneficial effects of EZH2 inhibitors on DSS-induced colitis occur in an adaptive immunity-independent manner.

Innate immunity plays a vital role in controlling intestinal inflammation during the early stage of IBD. Thus, we next analyzed the effect of GSK343 treatment on the infiltration of innate immune cell populations. Consistent with previous literature[29], colitis development was accompanied with heightened macrophage, DC, and eosinophil populations in the cLP; however, these increases was unaffected by GSK343 treatment. To our surprise, we observed increased percentages and numbers of MDSCs upon GSK343 treatment (Fig. 3a). This observation was strengthened by immunofluorescence staining showing significantly enhanced numbers of Gr-1-positive cells in the cLP after GSK343 administration (Fig. 3b). However, GSK343 did not appear to affect the immunosuppressive function of MDSCs,

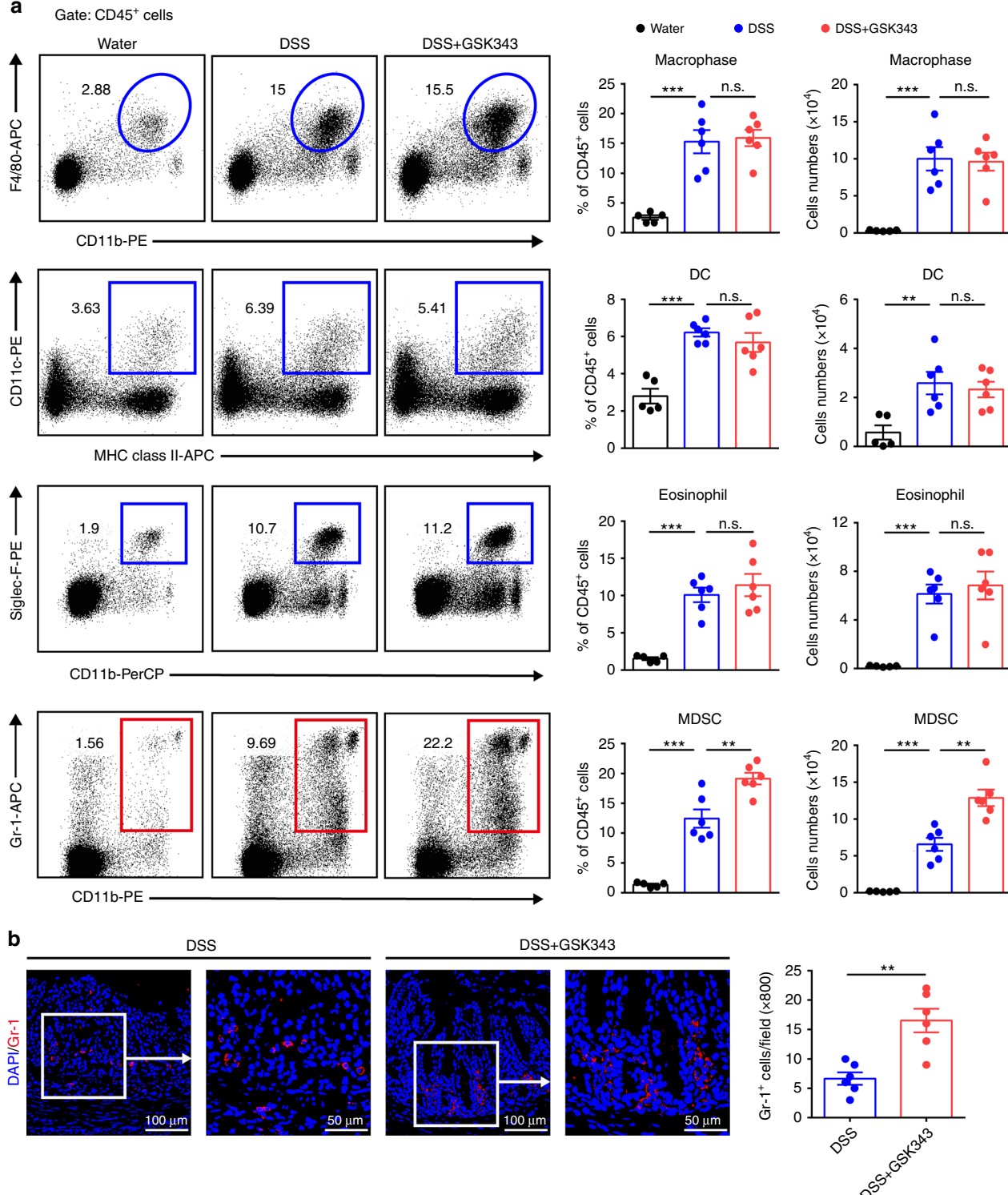

**Fig. 3** EZH2 inhibition increases colonic myeloid-derived suppressor cells (MDSCs). C57BL/6 mice were exposed to either 2.5% DSS-containing or regular drinking water alone for 5 days. DSS-treated mice were intravenously injected with either vehicle or GSK343 at days 0 and 3. **a** At day 5, the representative staining, percentage, and absolute number of macrophages (CD11b[+] F4/80[+]), dendritic cells (DCs) (major histocompatibility complex (MHC) class II[+] CD11c[+]), eosinophils (Siglec-F[+] Gr-1[-]), and MDSCs (CD11b[+] Gr-1[+]) in CD45[+] cells of the cLP were determined by flow cytometry ($n = 5$, Water; $n = 6$ for both DSS and DSS + GSK343). Dot plots are gated on CD45[+] cells. Numbers adjacent to the outlined areas indicate the percentage of the gated population in each group. **b** The expression of Gr-1 (red) in distal colon tissue sections from DSS-exposed mice treated with vehicle or GSK343 was detected by immunofluorescence staining. Magnified views of the marked areas are shown in the adjacent panels. The numbers of Gr-1[+] cells per field are enumerated on the right ($n = 6$ per group). Throughout, data are representative of three independent experiments. The statistical significance of differences was determined by one-way analysis of variance followed by Bonferroni post test (**a**) and unpaired two-tailed Student's *t*-test (**b**). *$P < 0.05$, **$P < 0.01$, ***$P < 0.001$, n.s. = not significant. Error bars indicate means ± SEM. Source data are provided as a Source Data file

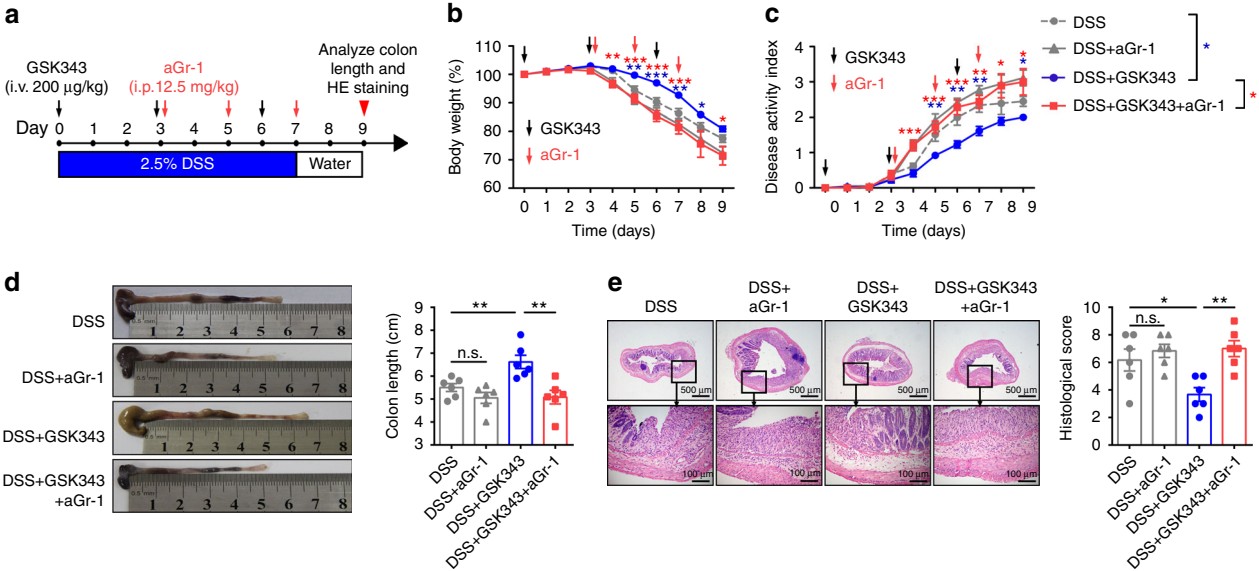

**Fig. 4** MDSC depletion reverses the colitis improvement rendered by GSK343. **a** Experimental design of DSS-induced colitis in C57BL/6 mice and treatment schedule. **b**, **c** Body weight (**b**) and disease activity index (**c**) of DSS-exposed mice that were treated with isotype control mAbs, anti-Gr-1, GSK343 plus isotype control mAbs, or GSK343 plus anti-Gr-1 ($n = 13$ per group). For statistical comparisons, blue asterisk (*) indicates DSS vs. DSS + GSK343 and red asterisk (*) indicates DSS + GSK343 vs. DSS + GSK343 + aGr-1. **d** Representative images of colons and colon length from the indicated treatment cohorts at day 9 after DSS treatment ($n = 6$ per group). **e** Representative H&E staining of distal colon cross-sections and corresponding histological scores at day 9 after initial DSS exposure ($n = 6$ per group). Throughout, data are representative of three independent experiments. The statistical significance of differences was determined by two-way analysis of variance with Bonferroni post test (**b**, **c**) and one-way analysis of variance followed by Bonferroni post test (**d**, **e**). *$P < 0.05$, **$P < 0.01$, ***$P < 0.001$, n.s. = not significant. The results are presented as the means ± SEM. Source data are provided as a Source Data file

because the colonic MDSC expression of ROS, arginase-1, and inducible nitric oxide synthase, known critical mediators of MDSC function[30], were unaltered upon GSK343 treatment (Supplementary Fig. 6). Collectively, these findings show that the inhibition of EZH2 activity leads to increased MDSC populations but no apparent change of MDSC function.

The observation that MDSC populations in the cLP were increased upon GSK343 treatment prompted us to investigate whether these cells are functionally important in mediating the beneficial effect of GSK343 on colitis. Based on previous studies[31–33], we employed an anti-Gr-1-specific monoclonal antibody (RB6–8C5 antibody) to deplete MDSCs (Fig. 4a). As reported, a single injection of anti-Gr-1 antibody effectively depleted MDSCs in the peripheral blood and cLP (Supplementary Fig. 7). Strikingly, depletion of MDSCs reversed the beneficial effects of GSK343 on colitis. The classic signs of colitis, including body weight loss (Fig. 4b), DAI (Fig. 4c), colon shortening (Fig. 4d), and pathology (Fig. 4e), all worsened in GSK343-treated mice upon anti-Gr-1 antibody administration. These results strongly suggest that the protective effects on colitis of EZH2 methyltransferase inhibition depend on elevated MDSCs in the cLP.

**EZH2 inhibits MDSC generation from HPCs**. As GSK343-treated mice exhibit elevated MDSCs in the cLP, we wondered whether GSK343 facilitates MDSC chemotaxis. To test this, we analyzed the expression of MDSC-related chemokines including CXCL1, CXCL2, CXCL5, and CXCL12 in inflamed colon tissues isolated from DSS-exposed mice treated with or without GSK343. These colonic MDSC-related chemokines were not perturbed by GSK343 treatment (Supplementary Fig. 8a), suggesting that increased MDSC numbers in the colon may not be due to increased cell chemotaxis. A second possible explanation is that GSK343 treatment enhances MDSCs generation. Strikingly, we

observed a substantially elevated percentage of MDSCs both in the bone marrow (BM) and peripheral blood following GSK343 treatment (Supplementary Fig. 8b), indicating that elevated MDSCs in the cLP may stem from their increased generation in the BM.

To further determine whether EZH2 inhibition leads to similar MDSC increases under steady state without apparent inflammation, we detected MDSC populations in healthy mice treated with different doses of GSK343. Indeed, GSK343 treatment resulted in a dose-independent increase in MDSC populations in the BM and peripheral blood of healthy animals (Supplementary Fig. 8c). Elevated MDSC percentages were also observed upon treatment with GSK126 (Supplementary Fig. 8d), indicating that blocking EZH2 methyltransferase activity causes an increase in MDSC populations in a cell-intrinsic manner independent of inflammation.

As GSK343 treatment leads to increased percentages of MDSCs in the BM, where MDSCs are generated, it is possible that inhibiting EZH2 activity may promote HPC differentiation into MDSCs. To test this, we first sorted Lin⁻Sca-1⁻ C-kit⁺ BM HPCs (Supplementary Fig. 9a) and treated them with granulocyte-macrophage colony-stimulating factor (GM-CSF) and IL-6 for 96 h, to allow MDSC induction in vitro (Fig. 5a). Strikingly, we found a gradual loss of EZH2 and H3K27Me3 expression (Fig. 5b), accompanied by a gradual expansion of CD11b⁺Gr-1⁺ MDSC populations (Fig. 5c), suggesting that loss of EZH2 activity may be responsible for MDSC generation. To further investigate this possibility, we analyzed the effect of GSK343 on MDSC development by treating cultured cells with GSK343 at multiple time points (Fig. 5a). Western blotting confirmed that 5 μM GSK343 specifically reduced H3K27Me3 levels (Fig. 5d). Notably, 5 μM GSK343 significantly enhanced the induction of MDSCs from HPCs (Fig. 5e) without apparently affecting their proliferation (Supplementary Fig. 9b and c), apoptosis (Supplementary

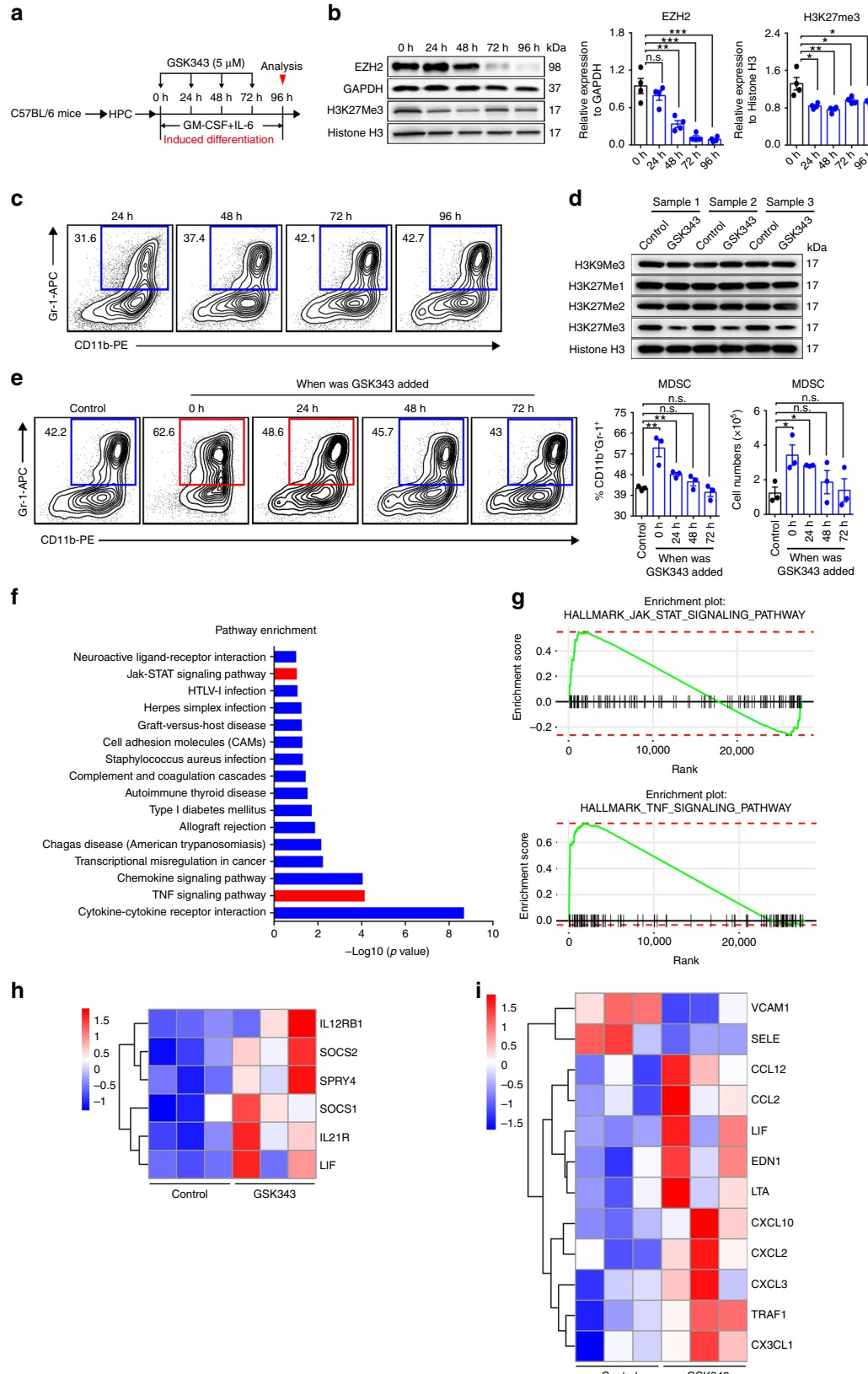

Fig. 9d), or the further differentiation of MDSCs (Supplementary Fig. 9e).

To uncover potential molecular mechanisms underlying the GSK343-mediated facilitation of MDSC induction, we performed RNA sequencing (RNA-seq) analysis of in vitro-induced HPCs treated with or without GSK343. We found that GSK343 activates the Jak-STAT and TNF signaling pathways, which are known to be involved in MDSC production[30,34] (Fig. 5f–i), indicating that GSK343 may facilitate HPC differentiation into MDSCs by activating Jak-STAT and TNF signaling. Based on these results,

**Fig. 5** Inhibition of EZH2 activity promotes MDSC generation from HPCs. **a–c** Sort-purified HPCs from mouse BM cells were cultured with GM-CSF and IL-6 to allow induction of MDSCs in vitro. **a** Schematic of in vitro MDSC differentiation from mouse HPCs and experimental design for GSK343 treatment. **b** Immunoblotting of EZH2 and H3K27Me3 in GM-CSF/IL-6-induced cells at the indicated time points of induction. GAPDH and Histone H3 were used as loading controls ($n = 4$ per group). The statistical significance of differences is vs. 0 h of cell induction. **c** Representative flow cytometry plots of MDSCs (CD11b$^+$ Gr-1$^+$) induced by GM-CSF/IL-6. Gated numbers represent the percentage of CD11b$^+$ Gr-1$^+$ cells. **d** Immunoblots of H3K9Me3, H3K27Me1, H3K27Me2, and H3K27Me3 protein following treatment with GSK343 for 72 h. Histone H3 was used as a loading control ($n = 3$ per group). **e** As described in **a**, MDSCs were induced from mouse HPCs, vehicle (control), or GSK343 was added at different induction times. The total cells were collected at 96 h and representative staining, frequencies, and absolute numbers of developing MDSCs were assessed by flow cytometry. Numbers adjacent to the outlined areas indicated the percentage of CD11b$^+$ Gr-1$^+$ MDSCs ($n = 3$ per group). The statistical significance of differences was vs. control. **f** Signaling pathway enrichment analysis was performed using Kyoto Encyclopedia of Genes and Genomes (KEGG). Significantly enriched (nominal $P < 0.05$) pathways in in vitro-induced HPCs treated with or without GSK343 are plotted by enrichment score ($n = 3$). **g** Enrichment plot of the HALLMARK Jak-STAT and TNF signaling pathways for the comparison between in vitro-induced HPCs treated with vehicle (control) and those treated with GSK343. **h** Heatmap displays the differentially expression of Jak-STAT signaling pathway genes in vitro-induced HPCs treated with vehicle (control) and GSK343. **i** Heatmap illustrating the differentially expressed genes of TNF signaling pathway in vitro-induced HPCs treated with vehicle (control) and GSK343. Data shown are mean ± SEM and are representative of three independent experiments. *$P < 0.05$, **$P < 0.01$ ***$P < 0.001$, n.s. = not significant (as determined by one-way analysis of variance followed by Bonferroni post test). Source data are provided as a Source Data file

we conclude that EZH2 inhibition increases MDSC generation, at least in part, by promoting the differentiation of HPCs into MDSCs.

## Discussion

IBDs are globally prevalent and a well-known cause of colorectal cancer without effective therapy. The reversibility of histone modifications makes them suitable targets for therapeutic intervention. Although EZH2 appears to function as a pivotal immune regulator, how EZH2 affects the progress of inflammatory and immune diseases such as IBD remains poorly understood. Here we demonstrated, for the first time, that interfering with EZH2 activity using low doses of pharmacological inhibitors ameliorates DSS-induced colitis. Specifically, therapeutically targeting EZH2 activity not only reduces colitis-related symptoms and extends the survival of colitic mice, but, even more remarkably, this therapeutic approach also delays CAC and reduces tumor burden. EZH2 inhibitors are currently being tested in human clinical trials for the treatment of various malignancies[24]. Thus, our findings broaden the potential clinical application of these inhibitors to the treatment of IBD. Furthermore, it is noteworthy that the dose used in our IBD models is <1% of that used in a previous cancer model[35], indicating that these inhibitor(s) may be more potent, safe, and economical in treating IBD.

A recent report showed that epithelial EZH2 is responsible for maintaining epithelial cell barrier integrity and homeostasis, suggesting a protective role against colitis[36]. In contrast, inhibiting EZH2 with GSK126 in vitro has been proposed to restore intestinal homeostasis by normalizing the Paneth cell population, indicating a disruptive role of EZH2 in intestinal homeostasis[37]. These seemingly contradictory conclusions reveal the complex and diverse role of EZH2 in IBD pathogenesis, because its expression on distinct cell types may exert opposing effects on the intestinal inflammation process. Moreover, prior conclusions regarding the role of EZH2 in intestinal epithelium are largely based on studies using epithelial-specific conditional knockout mice without considering the effect of EZH2 on immune cells. Nonetheless, when considering clinical implications, systemic effects must be taken into account. In our DSS model with systemic GSK343 treatment, both epithelial and immune cells are affected. Here we surprisingly found that GSK343 treatment led to increased MDSC production and reduced intestinal inflammation. These results suggest that DSS colitis is predominantly caused by tissue damage-triggered inflammation and relieving inflammation by EZH2 inhibition is sufficient to prevent colitis development, despite the fact that epithelial cells may be more sensitive to damage when EZH2 activity is disrupted.

Intriguingly, global inhibition of EZH2 protein expression with DZNep was recently reported to result in an activated, effector phenotype of Treg cells, with worsened DSS colitis[28]. We tested the same cell-surface markers in Treg cells as previously reported; however, none of these markers were perturbed by GSK343 treatment. In addition, GSK343 only caused slightly reduced Treg cell populations, which we later showed are not functionally important in mediating the beneficial effect of GSK343 on DSS colitis. The reason for these seemingly conflicting results may be that the drug dose we used was too low to affect adaptive immunity. Alternatively, T-cell regulation might require an enzyme activity-independent EZH2 function, which could not be evaluated using selective EZH2 enzyme inhibitors. Distinct from GSK343 or GSK126, DZNep is an $S$-adenosylhomocysteine hydrolase inhibitor, rather than a direct, on-target inhibitor of EZH2; therefore, it affects many histone methylation marks in addition to those modified directly by EZH2[38,39]. DZNep has also been reported to function independently of EZH2[40]. Thus, one cannot rule out that DZNep aggravates the colitic process through other, possibly EZH2-independent means. Moreover, DZNep inhibits EZH2 by downregulating EZH2 protein levels. Although the canonical function of EZH2 is gene repression through H3K27 methylation, EZH2 can also act independently of H3K27Me3[41]. For example, EZH2 interacts with and methylates STAT3, resulting in STAT3 activation[42]. Strikingly, STAT3 is crucial for MDSC expansion[43], which is important for restraining intestinal inflammation[44]. In fact, we have prophylactically treated DSS-induced colitis mice with high doses of DZNep, as reported[28], confirming the result that DZNep heightens intestinal inflammation reactivity (Supplementary Fig. 10a–e). Of note, we found comparable percentages and numbers of MDSCs upon DZNep treatment (Supplementary Fig. 10f), an effect likely due to the functional compromise between H3K27Me3-dependent and H3K27Me3-independent regulation. Thus, H3K27Me3-independent EZH2 activity may also contribute to DZNep-mediated exacerbation of DSS-induced colitis. In this regard, these results have important implications such that, when targeting EZH2 to treat IBD, we should inhibit EZH2 methyltransferase activity rather than impede its expression.

The finding that colitis development leads to a marked increase in MDSC populations is consistent with previous findings in animals and patients with IBD[8,9]. Surprisingly, such an expansion was further elevated by GSK343 treatment with a concomitant mitigation of colitis. We noted that deletion of MDSCs at an early phase of colitis abrogates the colitis improvement provided by GSK343 treatment, strongly indicating that MDSCs are functionally important in mediating the beneficial effect of GSK343

on colitis. Our observations are in agreement with previous studies, which suggest a protective role of MDSCs in restraining inflammation and promoting tissue repair during experimental IBD[31,44]. However, clinical data regarding the role of MDSCs in IBD remains scarce and more studies are needed to completely elucidate the functions of MDSCs in IBD patients and to explore their potential therapeutic benefit. MDSCs are a heterogeneous population consisting of granulocytic and monocytic subsets[30]. Currently, the role of each MDSC subset in intestinal inflammation remains controversial. Although evidence is available to support the beneficial role of each subset[10,45], MDSC subsets have also been reported to function as pro-inflammatory myeloid cells. Adoptively transferred Ly6C$^{high}$ monocytes can convert into pro-inflammatory DCs and further contribute to intestinal inflammation[46]. Moreover, it has been reported that colitis-induced immunosuppressive MDSCs may represent a permissive rather than suppressive mechanism for Th17 generation[47]. Therefore, how EZH2 inhibitors influence each MDSC subset and their detailed roles in the intestinal immune response requires further investigation.

EZH2 is involved in BM hematopoietic stem cell proliferation and differentiation[16,48], as well as regulation of immune cell lineage determination[49]. Previous studies have reported a rapidly downregulated EZH2 expression during hematopoietic stem cell differentiation[48]. Notably, our study verified this substantial downregulation of EZH2 together with H3K27Me3 during MDSC differentiation from HPCs in vitro, suggesting a possible involvement of reduced EZH2 activity during the generation of MDSCs. Indeed, we found that EZH2 inhibition promotes the generation of MDSCs from hematopoietic progenitors in a cell-intrinsic manner. MDSCs promote tumor progression by inhibiting anti-tumor immunity as well as directly stimulating tumor development by promoting tumor cell survival, invasion, metastasis, and angiogenesis[43,50]. Therefore, the efficacy of EZH2 enzymatic inhibitors for cancer treatment should be evaluated using immune-competent mouse models rather than relying on immune-deficient models as previously reported[35]. Jak-STAT and TNF signaling pathways have crucial functions in promoting MDSC generation. For instance, activation of Jak-STAT pathway leads to the activation of transcription factors/regulators including CCAAT-enhancer-binding protein-β and Myc to promote the proliferation and differentiation of myeloid progenitors to functional MDSCs[51,52]. In addition, TNF signaling is important in the induction, survival, and accumulation of immunosuppressive MDSCs[34,53,54]. We found that inhibiting EZH2 methyltransferase activity during HPC differentiation into MDSCs altered the genes controlling Jak-STAT and TNF signaling pathways, suggesting that EZH2 may regulate Jak-STAT and TNF signaling pathways through histone methyltransferase activity, hence modulating MDSC generation from HPCs. Further investigations are warranted to clarify the detailed molecular mechanisms. In sum, our findings not only uncover a novel role of EZH2 in regulating MDSC development, but also suggest that to achieve better anticancer outcomes, pharmacological inhibition of EZH2 in cancer therapy should be used in combination with MDSC depletion.

In conclusion, the current study demonstrates that targeting EZH2 methyltransferase activity with pharmacological inhibitors may represent a safe, effective, and economical therapeutic approach to treat intestinal inflammation. In addition, our findings also provide new insight into the contribution of EZH2 to the development of MDSCs and suggest that caution should be taken when considering pharmacological inhibition of EZH2 activity as an anticancer strategy, as this approach may hinder beneficial anticancer immunity by producing immunosuppressive MDSCs.

## Methods

**Mice**. C57BL/6, NOD/SCID mice were purchased from the Chinese Academy of Medical Sciences (Beijing, China). All mice were kept in laminar flow cabinets under a specific pathogen-free environment. Female animals were used for all studies and were 6–8 weeks of age at the start of the experiments. Mouse care and use were approved by the Third Military Medical University Institutional Animal Care and Use Committee. All experimental procedures with mice were performed in accordance and compliance with the regulations of the Laboratory Animal Welfare and Ethics Committee of the Third Military Medical University.

**Drugs**. GSK343 (Selleck Chemicals) was resuspended in anhydrous ethanol with moderate vortex and then diluted in phosphate-buffered saline (PBS) or in SFEM (STEMCELL Technologies) medium for in vivo and in vitro studies before use, respectively. GSK126 (Selleck Chemicals) was dissolved in 20% captisol with moderate vortex and then diluted in PBS. For all in vivo studies, GSK343 or GSK126 was provided by i.v. injection at a dose volume of 0.2 ml per 20 g body weight and equivalent amount of ethanol or captisol was injected in vehicle control mice, respectively.

**Induction of intestinal inflammation and treatments**. DSS colitis was induced by adding 2.5% DSS (molecular mass 36,000–50,000 Da; MP Biomedicals) to drinking water for 7 days, followed by normal drinking water for the remaining days. To induce enteropathy, mice were gavaged once daily with indomethacin (5 mg/kg in PBS for 5 days; Selleck Chemicals, Houston, TX, USA). In the preventive treatment, mice were i.v. injected with GSK343 or GSK126 every 3 days, starting from day 0 (DSS induction) until they were euthanized. For the therapeutic experiment, mice were i.v. injected with GSK343 every other day until they were killed. For the MDSC blockade experiment, 12.5 mg/kg anti-Gr-1 (RB6–8C5; BioXCell) or isotype control (RatIgG2b; BioXCell) was administered intraperitoneally (i.p.) every other day. We used the DAI to quantify colitis severity as previously described[55]. When we monitored the survival rate, mice received 3% DSS to induce intestinal inflammation and the natural death time of mice in the indicated groups within 15 days was recorded.

**Cytokine ELISA detection**. Proteins were extracted from distal colons of healthy and DSS-induced IBD mice treated with or without GSK343. The concentrations of IL-6, IL-1β, IL-17, and TNF-α were quantified by enzyme-linked immunosorbent assay following the manufacturer's instructions (Mlbio, Shanghai, China). The absorbance was determined at 450 nm using Varioskan Flash (Thermo Scientific, MA, USA).

**Histological analysis**. Briefly, a 1 cm segment of the distal colon was immediately fixed with 4% paraformaldehyde and embedded in paraffin, 3 μm sections were cut, and stained with hematoxylin and eosin. Histological scoring was performed in a blinded fashion by a pathologist, according to previously published criteria:[56] crypt architecture (normal, 0–severe crypt distortion with loss of entire crypts, 3), degree of inflammatory cell infiltration (normal, 0–dense inflammatory infiltrate, 3), muscle thickening (base of crypt sits on the muscularis mucosae, 0–marked muscle thickening, 3), crypt abscess (absent, 0–present, 1), and goblet cell depletion (absent, 0–present, 1). The total histologic score was derived by summing each individual score.

**Induction of colitis-associated colon cancer and treatments**. Modeling and analysis of colitis-associated tumorigenesis was performed according to a previously reported protocol[57]. Briefly, age-matched, co-caged C57BL/6 mice were given a single i.p. injection of the mutagen AOM (Sigma-Aldrich) at a dose of 10 mg/kg body weight (day 0). One week later, 2% DSS was given in drinking water for 7 days followed by 14 days of regular drinking water for a total of two cycles. During each cycle, mice were i.v. injected with either vehicle or GSK343 (200 μg/kg) every other day (for a total of five injections) when more than half of the mice showed colitis symptoms. Upon necropsy, the numbers and size of visual polyps in colons were measured using a digital caliper.

**Isolation of cLP cells**. Colons were processed by first incubating in 5 mM EDTA (Biosharp), 1 mM dithiothreitol (Beyotime) and 5% fetal calf serum in Ca/Mg-free Hanks' balanced salt solution (Gibco) at 37 °C for 20 min twice with shaking to remove intestinal epithelial cells. Tissues were then digested in 1 mg/mL collagenase D (Roche) and 0.5 mg/mL DNase I (Roche) in IMDM (HyClone) supplemented with 5% fetal bovine serum for 1 h at 37 °C with shaking. The supernatant was then filtered through 100 μm cell strainers and resuspended in 40% Percoll, followed by overlaying on top of the 80% fraction of the Percoll (Sigma-Aldrich). After centrifugation at 1000 × g for 20 min without using the brakes, the cLP cells were obtained from the interphase of the two different Percoll solutions.

**Flow cytometry**. For cell-surface antigen staining, cells were pre-incubated with Fc Receptors Blocking Reagent (Miltenyi Biotec) for 15 min at 4 °C before being stained with antibodies. Fixable Viability Dye eFluor® (eBioscience) was added to

exclude dead cells. The following mouse antibodies were used for staining: CD45 (FITC, Biolegend, clone 30-F11, dilution 1:100, lot 103108), CD11b (PE, Biolegend, clone M1/70, dilution 1:100, lot 101208; PerCP, Biolegend, clone M1/70, dilution 1:100, lot 101230), F4/80 (APC, Biolegend, clone BM8, dilution 1:100, lot 123116), CD11c (PE, BD Pharmingen™, clone HL3, dilution 1:100, lot 557401), MHC class II (APC, BioLegend, clone M5/114.1, dilution 1:100, lot 107614), Siglec-F (PE, BD Pharmingen™, clone E50–2440, dilution 1:100, lot 562068), Gr-1 (APC, BioLegend, clone RB6–8C5, dilution 1:150, lot 108412), CD4 (FITC, BioLegend, clone GK1.5, dilution 1:100, lot 100406; Pacific Blue, BioLegend, clone GK1.5, dilution 1:100, lot 100428), IL-17A (APC, eBioscience, clone eBio17B7, dilution 1:100, lot 17–7177–81), Foxp3 (APC, eBioscience, clone FJK-16s, dilution 1:100, lot 171–5773–82), CD25 (PE, Biolegend, clone PC61, dilution 1:100, lot 102008), GITR (FITC, Biolegend, clone BNI3, dilution 1:100, lot 369614), ICOS (APC/Cy7, Biolegend, clone C398.4A, dilution 1:100, lot 313529), CTLA-4 (PE/Cy7, BioLegend, clone BNI3, dilution 1:100, lot 369614), CD39 (PE, Biolegend, clone Duha59, dilution 1:100, lot 143803), CD73 (PerCP, Biolegend, clone TY/11.8, dilution 1:100, lot 127214), DCFH-DA (Beyotime, dilution 1:1000, lot S0033), Lineage (PerCP/Cy5.5, BD Pharmingen™, clone AA4.1, dilution 1:100, lot 561317), Sca-1 (FITC, Biolegend, clone D7, dilution 1:100, lot 108106), C-kit (PE, Biolegend, clone 2B8, dilution 1:100, lot 105808), PI (Biolegend, dilution 1:20, lot 421301), Annexin V (APC, Biolegend, dilution 1:20, lot 640941), and 7-AAD (Biolegend, dilution 1:20, lot 420403). Incorporation of BrdU was detected with a BrdU Flow Kit (APC, BD Pharmingen™, dilution 1:100, lot 559619) according to the manufacturer's protocol. Flow cytometry and cell sorting was performed using a BD FACSCalibur flow cytometer and a BD FACSAria™ II cell sorter, respectively. Data were analyzed using the FlowJo software.

**Immunofluorescence staining**. Samples were incubated with specific primary antibodies against Gr-1 (1:100 dilution, rat anti-mouse, R&D Systems) overnight at 4 °C. After being washed with PBS, tissues were stained with Cy3-conjugated anti-rat antibodies (1:200 dilution, Beyotime Biotechnology) for 30 min at 37 °C. The nuclei were counterstained with 4′,6-diamidino-2-phenylindole. Fluorescence emission was observed with an Olympus confocal microscope. The number of positive cells per field of view under ×800 magnification was counted and data were collected from five randomly selected fields.

**RNA extraction and quantitative RT-PCR analysis**. Total RNA of cells or distal colonic tissues was extracted using Trizol (Invitrogen) or RNAqueous-Micro Kit (Life Technologies) according to the manufacturer's instructions. High-fidelity cDNA was generated from each RNA sample with a cDNA Reverse Transcription Kit (Takara). Quantitative reverse transcriptase-PCR was performed using a SYBR Kit (Takara) according to the manufacturer's protocol. We used the $2^{-\Delta\Delta Ct}$ quantification method with mouse glyceraldehyde 3-phosphate dehydrogenase (GAPDH) as an endogenous control. The primer sequences are listed in Supplementary Table 1.

**Generation of MDSCs from sorted HPCs and treatments**. For HPC isolation, lineage-negative cells were sorted from the BM of C57BL/6 mice followed by staining with Sca-1 (FITC, Biolegend, clone D7, dilution 1:100, lot 108106) and c-kit (PE, Biolegend, clone 2B8, dilution 1:100, lot 105808) antibodies. To differentiate HPCs into MDSCs, $2 \times 10^5$ Lin$^-$Sca-1$^-$C-kit$^+$ HPCs were placed in each well of 24-well plates and cultured in SFEM (STEMCELL Technologies) medium containing GM-CSF (10 ng/mL; PeproTech) and IL-6 (10 ng/ml; PeproTech). At different time points during culturing, 5 μM GSK343 (Selleck Chemicals) was added to the culture system and the newly generated CD11b$^+$Gr-1$^+$ cells were analyzed at 96 h.

**Western blotting**. Total protein was extracted with RIPA supplemented with 1% phenylmethylsulfonyl fluoride (Beyotime Biotechnology). Protein concentrations were then tested with a BCA kit (Beyotime Biotechnology). Primary antibodies rabbit anti-mouse EZH2 (1:1000 dilution, Cell Signaling Technology), rabbit anti-mouse GAPDH (1:1000 dilution, Beyotime Biotechnology), rabbit anti-mouse H3K27me3 (1:2000 dilution, Cell Signaling Technology), rabbit anti-mouse Histone H3 (1:1000 dilution, Cell Signaling Technology), rabbit anti-mouse H3K9Me3 (1:1000 dilution, Abcam), rabbit anti-mouse H3K27Me1 (1:1000 dilution, Multi Sciences), and rabbit anti-mouse H3K27Me2 (1:1000 dilution, Multi Sciences) were used to incubate polyvinylidene difluoride membrane at 4 °C overnight. The membranes were then incubated with goat anti-rabbit secondary antibody (1:5000, Beyotime Biotechnology). Images were visualized with a chemiluminescence detection system. All western blotting images were cropped to optimize clarity and presentation. Uncropped and unprocessed scans of the representative blots in this manuscript are provided in the Source Data file.

**RNA sequencing**. Sort-purified HPCs were treated with vehicle (control) or 5 μM GSK343 for 24 h in SFEM medium containing GM-CSF and IL-6. Total RNA from induced cells was collected using an RNAqueous-Micro Kit (Invitrogen). Samples were prepared for sequencing using a TrueLib mRNA Library Prep Kit for Illumina (ExCell Bio). RNA-seq libraries of these RNA samples were constructed according to standard Illumina protocols and sequenced using an Illumina HiSeq

2000 sequencer. Transcript abundance was calculated and normalized in fragments per kilobase of transcript per million mapped reads from the raw RNA-seq data. The Kyoto Encyclopedia of Genes and Genomes was used to interpret gene expression profiles.

**Statistical analysis**. All experiments were independently performed three times. Data are presented as the mean ± SEM. Differences between groups were evaluated with an unpaired two-tailed Student's $t$-test or by analysis of variance followed by Bonferroni post test. Log-rank test was used to evaluate survival difference. Statistical significance was defined as $P < 0.05$.

**Reporting summary**. Further information on research design is available in the Nature Research Reporting Summary linked to this article.

## Data availability

RNA-seq datasets can be accessed with the BioProject ID number PRJNA516073 at the NCBI BioProject database. Other data that support the findings of this study are available from the corresponding author upon reasonable request. The source data underlying Figs. 1b-g, 2b–f, 3a-b, 4b-e, 5b,d,e and Supplementary Figs 1b-c, 2b-f, 3b-d, 4b,e,h, 5a,d-g, 6a-b, 7a-b, 8a-d, 9b-e and 10b-f are provided as a Source Data file.

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

## Acknowledgements

This work was supported by the National Natural Science Foundation of China (Number 81222031 to B.Z., Number 81773031 to H.L., and Number 81802865 to Z.W.).

## Author contributions

J.Z.: acquisition of data, analysis and interpretation of data, and drafting of the manuscript with comments from all authors. S.H. and Z.W.: acquisition of data, analysis and interpretation of data. J.H., L.X., and X.T.: acquisition of data. Y.Y.W., Q.L., and A.L.J.S.: study concept and design, and drafting of the manuscript. H.L.: study concept and design, analysis and interpretation of the data, and study supervision. B.Z.: study concept and design, study supervision, obtained financial support and administrative support, critical revision of the manuscript for important intellectual content. All authors have read and approved the manuscript for publication.

## Additional information

**Competing interests:** The authors declare no competing interests.

