## [Peer Review File · Nature Communications]

Reviewers' Comments:

Reviewer #1:

Remarks to the Author:

Zhou Jie et al. describe the contribution of EZH2 histone methyltransferase inhibitors, GSK343 and GSK126, to reducing its methyltransferase activity, resulting in the restrained inflammation in experimental colitis induced by means of water-fed dextran sodium sulfate (DSS). Since EZH2 is originally reported not only to be as a therapeutic target for lymphoma (Nature 2012), but also to serve as an epigenetic determinant in the model of experimental colitis (JBC 2017, PNAS 2017), the rationale why the authors have initiated this study is logical, but the effectiveness of EZH2 histone methyltransferase inhibitors on the improvement of mouse colitis in this study is not surprising. What is most striking in the present study is to find the increased number of immunosuppressive myeloid-derived suppressor cells (MDSCs) when treated with the inhibitors. Considering EZH2 is an epigenetic writer of trimethylation of histone H3K27, although the pathophysiological works using mice are well done to investigate the effect of inhibitors on the attenuation of colitis, it is too preliminary to examine the action of inhibitors on epigenetic regulation in MDSCs. These studies are required not only for examining the mechanistic insight of gene expression alteration, but also for evaluating the risk and benefit of inhibitors for its use to the treatment of inflammatory bowel disease (IBD) as a clinical trial.

Major concerns:

- 1) In Figure 5D (at 72 hours later with treatment of GSK343), it is very important to evaluate the change in histone methylation, because it is an inhibitor for EZH2. Figure 5B demonstrates clear reduction of EZH2 protein accumulation (or expression) in MDSCs derived from BM HPCs that are sorted as Lin⁻ Sca-1⁻ C-kit⁺ treated with GM-CSF and IL-6 for 96 hours. The authors should examine the changes in methylation levels of H3K27Me₃, and H3K27Me₂ and H3K27Me₁ as controls in a time-course dependent manner.
- 2) Furthermore, if possible, examining other sites for methylation of histones will be given useful information regarding the specificity of EZH2, because the title includes METHYLTRANSFERASE ACTIVITY.
- 3) In order to avoid the bias to pick up candidates of your interest for checking the changes in immune related genes, non-bias methods are required to argue the effectiveness of inhibitors. RNA sequence or micro array analysis will be available using MDSCs. Once candidate genes are chosen by non-bias methods, it should be examined whether or not they are targets of EZH2.

Minor concerns:

- 4) The word, DSS, is appearing at first line 85, and it is also seen at line 95. However, its full spelling comes at line 103. The full spelling, dextran sodium sulfate, and its abbreviation, DSS should be first at line 85.
- 5) At line 298, the two words, Thus, Therefore, are duplicated. Please choose one.
- 6) At line 300, target should be targetS, if the present form is used.
- 7) At line 305, prophylactic should be prophylacticALLY, as seen at line 187.
- 8) In Figures 1 ~ 5, A, B, C, and so on are used to indicate each figures or chart by capital letters, but in figure legends, lower cases are used at the corresponding parts for the explanation.

Reviewer #2:

Remarks to the Author:

The authors perform an evaluation of EzH2 inhibitors in the treatment of Chemically induced colitis. They demonstrate that chemically induced colitis can be inhibited with these compounds. The authors then go further to explain the beneficial effects by invoking the increase in MDSCs as the potential target of inhibition of the methyltransferase. Overall I am enthusiastic about this submission. I had many concerns that were raised through the reading of the paper and much of this was alleviated by the data presented.

Persistent concerns:

1) can the authors comment on the specificity of these compounds? They have a significantly different profile of effects compared to other EzH2 inhibitors.

2) I remain concerned that the studies here show a completely different phenotype on DSS induced colitis as compared to the paper in PNAS by Liu et al 2017. Although they were evaluating Epithelial cell integrity, which is the primary target of DSS, they demonstrate that over expression of EzH2 protects against DSS induced colitis. I understand that inhibition of function would be different than deletion of the gene but the overexpression does not support that this difference is the explanation of the differential effects between the PNAS paper and this submission

3) Has there been any attempt to use a more immune mediated model of colitis as DSS is considered more of an epithelial damage model and immune mediated.

4) How do the authors explain the identified decrease in EzH2 expression in IBD patients? how does it corroborate this data or submission.

5) The effect on Tregs of ezh2 loss or inhibition is to significantly alter Treg function can the authors better explain why this ezh2 inhibitor does not have a similar effect?

6) While the NOD/SCID experiments are important to demonstrate an adaptive immune independent mechanism could this not also be related to epithelial cell function or integrity.

Minor point:

Some work on language and grammar is need for the submission

6)

RESPONSES TO REVIEWERS

We would like to express our sincere thanks to both reviewers for their critical and constructive comments. We have performed substantial additional analyses to address their concerns. Below, we respond point-by-point to each of their comments and criticisms. We feel that their comments have helped us significantly improve and strengthen the manuscript and clarify key issues.

Response to Reviewer #1:

Zhou Jie et al. describe the contribution of EZH2 histone methyltransferase inhibitors, GSK343 and GSK126, to reducing its methyltransferase activity, resulting in the restrained inflammation in experimental colitis induced by means of water-fed dextran sodium sulfate (DSS). Since EZH2 is originally reported not only to be as a therapeutic target for lymphoma (Nature 2012), but also to serve as an epigenetic determinant in the model of experimental colitis (JBC 2017, PNAS 2017), the rationale why the authors have initiated this study is logical, but the effectiveness of EZH2 histone methyltransferase inhibitors on the improvement of mouse colitis in this study is not surprising. What is most striking in the present study is to find the increased number of immunosuppressive myeloid-derived suppressor cells (MDSCs) when treated with the inhibitors. Considering EZH2 is an epigenetic writer of trimethylation of histone H3K27, although the pathophysiological works using mice are well done to investigate the effect of inhibitors on the attenuation of colitis, it is too preliminary to examine the action of inhibitors on epigenetic regulation in MDSCs. These studies are required not only for examining the mechanistic insight of gene expression alteration, but also for evaluating the risk and benefit of inhibitors for its use to the treatment of inflammatory bowel disease (IBD) as a clinical trial.

We thank Reviewer #1 for his/her enthusiasm for our study. Based on Reviewer #1's suggestions, we have performed additional experiments to address these concerns as described below.

Major concerns:

1) In Figure 5D (at 72 hours later with treatment of GSK343), it is very important to evaluate the change in histone methylation, because it is an inhibitor for EZH2. Figure 5B demonstrates clear reduction of EZH2 protein accumulation (or expression) in MDSCs derived from BM HPCs that are sorted as Lin⁻ Sca-1⁻ C-kit⁺ treated with GM-CSF and IL-6 for 96 hours. The authors should examine the changes in methylation levels of H3K27Me₃, and H3K27Me₂ and H3K27Me₁ as controls in a time-course dependent manner.

We thank Reviewer #1 for this constructive comment to improve this study. Per this suggestion, we examined the levels of H3K27Me₃ in GM-CSF/IL-6 induced HPCs at different time points during induction. Consistent with our previous results showing

decreased EZH2 expression during MDSC differentiation from HPCs, we also observed a reduction in H3K27Me3 levels in a time-dependent manner (**Response Figure 1**). We have added these results to the revised manuscript as new Figure 5b. (Please see the changes in line 255, 257-258, page 12 in the revised manuscript, also Fig 5b. and the corresponding figure legends.)

Response Figure 1 Immunoblot of EZH2 and H3K27Me3 in GM-CSF/IL-6-induced cells at the indicated time points of induction. GAPDH and Histone H3 were used as loading controls (n = 4 per group). Statistical analysis is compared to 0 h of cell induction.

2) Furthermore, if possible, examining other sites for methylation of histones will be given useful information regarding the specificity of EZH2, because the title includes METHYLTRANSFERASE ACTIVITY.

We agree with this comment. Per Reviewer #1's suggestion, upon GSK343 treatment for 72 h, we measured the expression of other sites of histone methylation, including H3K9Me3, H3K27Me1 and H3K27Me2, to determine the specificity of EZH2 methyltransferase inhibition. Consistent with the literature^{1, 2}, inhibition of EZH2 methyltransferase activity with GSK343 specifically decreased H3K27 trimethylation levels without affecting other histone methylation sites (**Response Figure 2**). We have added these results in the revised manuscript as new Figure 5d. (Please see the changes in line 260, page 12; line 356-357, page 16; line 514-516, page 23 in the revised manuscript, also Fig 5d. and the corresponding figure legends.)

Response Figure 2 Immunoblots of H3K9Me3, H3K27Me1, H3K27Me2 and H3K27Me3 protein levels following treatment with GSK343 for 72 h. Histone H3 was used as a loading control (n = 3 per group).

3) In order to avoid the bias to pick up candidates of your interest for checking the changes in immune related genes, non-bias methods are required to argue the effectiveness of inhibitors. RNA sequence or micro array analysis will be available using MDSCs. Once candidate genes are chosen by non-bias methods, it should be examined whether or not they are targets of EZH2.

We thank Reviewer #1 for this excellent suggestion. We performed RNA sequence analysis of *in vitro*-induced HPCs treated with or without GSK343 to search for the potential mechanism that may be responsible for the GSK343-mediated promotion of MDSC generation. Then, we used the Kyoto Encyclopedia of Genes and Genomes (KEGG) for signaling pathway enrichment. The results show that GSK343 activates the Jak-STAT and TNF signaling pathway which are known to be involved in MDSC production^{3, 4} (**Response Figure 3**). These data indicate that GSK343 may promote HPC to MDSC differentiation by activating Jak-STAT and TNF signaling. We have added these results in the revised manuscript as new Figure 5f. (Please see the changes in line 265-269, page 12; line 270, page 13 and line 521-531, page 24 in the revised manuscript, also Fig. 5f. and the corresponding figure legends.)

Response Figure 3 Signaling pathway enrichment analysis was performed using Kyoto Encyclopedia of Genes and Genomes (KEGG). Significantly enriched (nominal $P < 0.05$) pathways in *in vitro*-induced HPCs treated with or without GSK343 are plotted by enrichment score (n = 3).

Minor concerns:

4) The word, DSS, is appearing at first line 85, and it is also seen at line 95. However, its full spelling comes at line 103. The full spelling, dextran sodium sulfate, and its abbreviation, DSS should be first at line 85.

We appreciate Reviewer #1's careful reading of our manuscript. We have made these corrections in the revised manuscript. (Please see the changes in line 88, page 4 and line 106, page 5 in the revised manuscript) .

5) At line 298, the two words, Thus, Therefore, are duplicated. Please choose one.

We apologize for this mistake and have corrected this in the revised manuscript (Please see the changes in line 318, page 15 in the revised manuscript).

6) At line 300, target should be targetS, if the present form is used.

We are grateful to Reviewer #1 for the improvements made to this manuscript. We have made this correction in the revised manuscript (Please see the changes in line 319, page 15 in the revised manuscript) .

7) At line 305, prophylactic should be prophylacticALLY, as seen at line 187.

Per Reviewer #1's suggestion, we have changed the word "prophylactic" to "prophylactically." (Please see the changes in line 324, page 15 in the revised manuscript) .

8) In Figures 1 ~ 5, A, B, C, and so on are used to indicate each figures or chart by capital letters, but in figure legends, lower cases are used at the corresponding parts for the explanation.

We apologize for these mistakes and have made the corrections in the revised manuscript (Please see the changes in Figures 1 ~ 5 and the corresponding figure legends)

Response to Reviewer #2:

The authors perform an evaluation of EZH2 inhibitors in the treatment of Chemically induced colitis. They demonstrate that chemically induced colitis can be inhibited with these compounds. The authors then go further to explain the beneficial effects by invoking the increase in MDSCs as the potential target of inhibition of the methyltransferase. Overall, I am enthusiastic about this submission. I had many concerns that were raised through the reading of the paper and much of this was alleviated by the data presented.

We thank Reviewer #2 for his/her enthusiasm for our study.

Persistent concerns:

1) can the authors comment on the specificity of these compounds? They have a significantly different profile of effects compared to other EZH2 inhibitors.

We thank Reviewer #2 for pointing out this issue, which is an important one. To determine the specificity of these compounds, we first tested the levels of H3K27 trimethylation following treatment with GSK343 because all drugs used in our study target the methyltransferase activity of EZH2, which is manifested as reducing H3K27me3 levels, rather than downregulating EZH2 expression. H3K27Me1 and H3K27Me2 protein were used as controls to further strengthen and clarify the conclusions since they are not generated by EZH2 but by other histone methyltransferases such as EHMT2. Consistent with the literature,² GSK343 treatment did not affect the expression of EZH2 or levels of H3K27Me1 and H3K27Me2, but instead specifically decreased H3K27me3 levels (**Response Figure 4a**).

In addition, we compared EZH2 and H3K27Me levels after treatment with DZNep, another widely used EZH2 inhibitors that inhibits S-adenosyl-homocysteine hydrolase, resulting in reduced EZH2 expression. As shown in **Response Figure 4b**, distinct from GSK343, DZNep treatment effectively reduced the expression of EZH2 as well as H3K27Me3. Based on these results, we conclude that the selective EZH2 methyltransferase inhibitor GSK343 specifically inhibits EZH2 activity, and is distinct from other EZH2 inhibitors that act by suppressing EZH2 expression.

Response Figure 4 (a) Immunoblots of EZH2, H3K27Me1, H3K27Me2 and H3K27Me3 protein following treatment with GSK343 for 72 h. GAPDH and Histone H3 were used as loading controls ($n = 3$ per group). **(b)** Immunoblots of EZH2, H3K27Me1, H3K27Me2 and H3K27Me3 protein following treatment with DZNep for 72 h. GAPDH and Histone H3 were used as loading controls ($n = 3$ per group).

2) I remain concerned that the studies here show a completely different phenotype on DSS induced colitis as compared to the paper in PNAS by Liu et al 2017. Although they were evaluating Epithelial cell integrity, which is the primary target of DSS, they demonstrate that over expression of EZH2 protects against DSS induced colitis. I understand that inhibition of function would be different than deletion of the gene but the overexpression does not support that this difference is the explanation of the differential effects between the PNAS paper and this submission.

We thank Reviewer #2 for raising this issue. EZH2 is known to be expressed in various cell types and function in a cell type-specific manner. The cited study showed that EZH2 is required to promote epithelial integrity without considering the effects of EZH2 on immune cell populations.⁵ Indeed, as Reviewer #2 predicted, we initially hypothesized that GSK343 might worsen colitis due to reduced intestinal epithelial integrity. However, in our DSS model with systemic GSK343 treatment, both epithelial and immune cell are affected. We surprisingly found that GSK343 treatment led to increased MDSCs and reduced intestinal inflammation. These results suggest that DSS colitis is predominantly caused by tissue damage-triggered inflammation and that relieved inflammation from EZH2 inhibition is sufficient to prevent colitis development even though epithelial cells may be more sensitive to damage when EZH2 activity is disrupted. We have further elaborated on this point in our revised discussion section. Please see the changes in line 295-304, page 14 in the revised manuscript.

3) Has there been any attempt to use a more immune mediated model of colitis as DSS is considered more of an epithelial damage model and immune mediated.

Per Reviewer #2's suggestion, we used another intestinal inflammation model, indomethacin-induced enteropathy, to observe the effect of EZH2 inhibition on intestinal inflammation. This model is widely used, mimics the clinical pathology of NSAID enteropathy, and is characterized by immune cell infiltration^{6,7}. Similar to our observations in the DSS model, GSK343 treatment also significantly reduced the severity of indomethacin-induced enteropathy, as evidenced by decreased weight loss, colonic shortening, and histological damage (**Response Figure 5a-d**). We have added these results in the revised manuscript as new Supplementary Fig. 2. Please see the changes in line 138-144, page 7 and line 398-399, page 18 in the revised manuscript, also Supplementary Fig. 2 and the corresponding figure legends.

Response Figure 5 Inhibition of EZH2 activity reduces the severity of indomethacin-induced enteropathy. (a) Flow diagram for indomethacin-induced acute enteropathy and GSK343 administration. (b) Body weights of mice that received regular drinking water alone (“Water” group; black line) or 5 mg/kg indomethacin (“NSAID” group; blue line) or 5 mg/kg indomethacin combined with GSK343 injection (“NSAID+GSK343” group; red line) ($n = 6$ per group). Asterisk (*) indicates “NSAID” group versus “NSAID+GSK343” group. (c, d) Intestine length (c), representative H&E staining of small intestine sections and corresponding histological scores (d) at day 5 after indomethacin administration ($n = 6$ per group). (b-d) Data are representative of three independent experiments. The statistical significance of differences was determined by two-way analysis of variance with Bonferroni post-test (b) and one-way analysis of variance followed by Bonferroni post-test (c, d). * $P < 0.05$, ** $P < 0.01$, *** $P < 0.001$. Error bars indicate means \pm SEM.

4) How do the authors explain the identified decrease in EZH2 expression in IBD patients? how does it corroborate this data or submission.

We thank Reviewer #2 for raising this question. Recently, a compelling study published in PNAS showed that EZH2 reduction in colorectal epithelium is associated with the pathogenesis of IBD⁵. However, in our study, we centered our investigations on EZH2’s role in immune regulation. Although our drugs may also affect intestinal epithelial cells, by using multiple mouse models and MDSC deletion experiments, we determined that systemic EZH2 activity inhibition effectively relieves intestinal inflammation to prevent the progression of experimental IBD. Thus, promoting MDSC production from hematopoietic progenitor cells is an important mechanism in IBD.

5) The effect on Tregs of ezh2 loss or inhibition is to significantly alter Treg function can the authors better explain why this ezh2 inhibitor does not have a similar effect?

We thank Reviewer #2 for raising this issue. We propose the following possible explanations for this phenomenon. First, as Reviewer #2 stated, Previous studies have shown that conditional EZH2 knockout in FOXP3-expressing cells or systemic EZH2 inhibition with DZNep significantly alters the phenotype and function of Tregs.⁵ In fact, both gene deletion and DZNep treatment results in down-regulation of EZH2 expression. Although the canonical function of EZH2 is repressing gene expression through methylation of H3K27, EZH2 also acts in other ways.⁶ For example, eEZH2 can interact with and methylate substrates other than H3K27, such as STAT3 and ROR α . Additionally, EZH2 has methylase-independent functions. Thus, inhibition of EZH2's methyltransferase activity is distinct from inhibition of its expression, thereby leading to differential effects. Second, the drug dose used in our models is much lower than that used with DZNep by Olga et al., which may also contribute to these seemingly inconsistent results.

6) While the NOD/SCID experiments are important to demonstrate an adaptive immune independent mechanism could this not also be related to epithelial cell function or integrity.

We thank Reviewer #2 for raising this issue. We agree that NOD/SCID experiments cannot rule out the effect of EZH2 on epithelial cell function or integrity but only demonstrate that adaptive immune cells are not required to mediate the phenotype. It is true that inhibition of EZH2 activity may also affect intestinal epithelial cell function or integrity; however, this is not the focus of our article. Moreover, the results in our study obtained using multiple mouse models and our MDSC depletion experiments suggest that increased MDSCs caused by EZH2 activity inhibition are an important mechanism and sufficient to prevent colitis development despite the possibility that intestinal epithelial cells are more sensitive to damage.

Minor point:

Some work on language and grammar is need for the submission

The revised manuscript has been edited as requested. (Please see our revised manuscript).

Reference

1. Fillmore CM, *et al.* EZH2 inhibition sensitizes BRG1 and EGFR mutant lung tumours to Topoll inhibitors. *Nature* **520**, 239-242 (2015).
2. Bitler BG, *et al.* Synthetic lethality by targeting EZH2 methyltransferase activity in ARID1A-mutated cancers. *Nature medicine* **21**, 231-238 (2015).
3. Gabrilovich DI, Nagaraj S. Myeloid-derived suppressor cells as regulators of the immune system. *Nature reviews Immunology* **9**, 162-174 (2009).

4. Sade-Feldman M, Kanterman J, Ish-Shalom E, Elnekave M, Horwitz E, Baniyash M. Tumor necrosis factor-alpha blocks differentiation and enhances suppressive activity of immature myeloid cells during chronic inflammation. *Immunity* **38**, 541-554 (2013).
5. Liu Y, *et al.* Epithelial EZH2 serves as an epigenetic determinant in experimental colitis by inhibiting TNFalpha-mediated inflammation and apoptosis. *Proceedings of the National Academy of Sciences of the United States of America* **114**, E3796-E3805 (2017).
6. Higashimori A, *et al.* Mechanisms of NLRP3 inflammasome activation and its role in NSAID-induced enteropathy. *Mucosal immunology* **9**, 659-668 (2016).
7. Whitfield-Cargile CM, *et al.* The microbiota-derived metabolite indole decreases mucosal inflammation and injury in a murine model of NSAID enteropathy. *Gut microbes* **7**, 246-261 (2016).

Reviewers' Comments:

Reviewer #1:

Remarks to the Author:

Jie Zhou et al. improved the revised manuscript by conducting new experiments including RNA sequencing. They try to address the questions to reviewers and appropriately accommodate them in the revised one except for the results of RNA sequencing. Although the authors identified JAK-STAT and TNF signaling pathways from the results of enrichment analysis, it is only shown in Fig. 5f. The authors should sufficiently describe data of the change in RNA accumulation and finally discuss regarding how JNK-STAT and TNF signaling pathways connect to the regulation of histone modification mediated by EZH2.

Reviewer #2:

None

Dear Reviewer,

Thank you for your critical and constructive comments to improve the study and manuscript. We are pleased that you find our revised manuscript have addressed most of the reviewers' concerns satisfactorily. We understand that the manuscript still has some concerns. To address your comments, we have further re-analyzed data of the change in RNA accumulation and revised the manuscript accordingly. Attached below, please find the detailed point-by-point response to your comments. We hope that your concerns have been addressed satisfactorily in the current revised version.

Thank you very much and best regards.

RESPONSES TO REVIEWER

Reviewer #1 (Remarks to the Author):

Jie Zhou et al. improved the revised manuscript by conducting new experiments including RNA sequencing. They try to address the questions to reviewers and appropriately accommodate them in the revised one except for the results of RNA sequencing. Although the authors identified JAK-STAT and TNF signaling pathways from the results of enrichment analysis, it is only shown in Fig. 5f. The authors should sufficiently describe data of the change in RNA accumulation and finally discuss regarding how JNK-STAT and TNF signaling pathways connect to the regulation of histone modification mediated by EZH2.

Response:

We realized that this reviewer raised an important issue. Thanks for his/her good suggestions to improve our manuscript. Per Reviewer #1's suggestion, we re-analyzed the data of changes in RNA accumulation when HPCs exposed to GSK343 and showed the differentially expressed genes identified in Jak-STAT and TNF signaling pathways accordingly (**Response Figure 1**). As reported, both signaling pathways have several known crucial functions in promoting MDSC production. For instance, activation of Jak-STAT pathway leads to the activation of transcription factors/regulators like CCAAT-enhancer-binding protein beta (C/EBP β), Myc, which are well-known key positive regulators for the proliferation and differentiation of myeloid progenitors to functional MDSCs^{1,2}. TNF signaling pathways also play an important role in the induction, survival and accumulation of immunosuppressive MDSCs^{3,4,5}. Our results indicate that inhibition of EZH2 methyltransferase activity activates Jak-STAT and TNF signaling pathways, suggesting that EZH2 may regulate Jak-STAT and TNF signaling pathways through histone methyltransferase activity hence modulating MDSC generation from HPCs. Further investigations are still needed to clarify the detailed molecular mechanisms. We have added these results in the revised manuscript as new Figure 5g-i and further elaborated on this point in our revised discussion section. Please see the changes in line 363-374, page 17 in the revised manuscript, also Figure 5g-i and the corresponding figure legends.

Response Figure 1 (a) Enrichment plot of the HALLMARK Jak-STAT and TNF signaling pathways for the comparison between *in vitro*-induced HPCs treated with vehicle (control) and those treated with GSK343. **(b)** Heatmap displays the differential expression of Jak-STAT signaling pathway genes in *in vitro*-induced HPCs treated with vehicle (control) and GSK343. **(c)** Heatmap illustrating the differentially expressed genes of TNF signaling pathway in *in vitro*-induced HPCs treated with vehicle (control) and GSK343.

Reference

1. Veglia F, Perego M, Gabrilovich D. Myeloid-derived suppressor cells coming of age. *Nat Immunol* **19**, 108-119 (2018).
2. Trikha P, Carson WE, 3rd. Signaling pathways involved in MDSC regulation. *Biochimica et biophysica acta* **1846**, 55-65 (2014).
3. Zhao X, et al. TNF signaling drives myeloid-derived suppressor cell accumulation. *The Journal of clinical investigation* **122**, 4094-4104 (2012).
4. Sade-Feldman M, Kanterman J, Ish-Shalom E, Elnekave M, Horwitz E, Baniyash M. Tumor necrosis factor-alpha blocks differentiation and enhances suppressive activity of immature myeloid cells during chronic inflammation. *Immunity* **38**, 541-554 (2013).

5. Atrekhany KS, *et al.* TNF Neutralization Results in the Delay of Transplantable Tumor Growth and Reduced MDSC Accumulation. *Frontiers in immunology* **7**, 147 (2016).